# Trends of inequality in DPT3 immunization services utilization in Ethiopia and its determinant factors: Evidence from Ethiopian demographic and health surveys, 2000–2019

**Hailu Fekadu**[1]*, **Wubegzier Mekonnen**[2], **Aynalem Adugna**[3], **Helmut Kloos**[4],
**Damen Hailemariam**[2]

1 Department of Public Health, Arsi University College of Health Science, Assela, Ethiopia, 2 School of Public Health, Addis Ababa University College of Health Science, Addis Ababa, Ethiopia, 3 Department of Geography, Planning and Environmental Sonoma state University, Sonoma, California, San Francisco, United States of America, 4 Department of Epidemiology and Biostatistics, University of California, San Francisco, California, United States of America

* hailufekadu18@yahoo.com

**Data Availability Statement:** All relevant data are available within the paper and its Supporting Information files.

## Abstract

### Background

Low levels of diphtheria, tetanus toxoid, pertussis (DPT3) immunization services utilization and high deaths among under five children are concentrated in economically and socially disadvantaged groups, especially in low and middle-income countries, including Ethiopia. Hence, the aim of this study is to assess levels and trends in DPT3 immunization services utilization in Ethiopia and identify inequalities.

### Methods

This study used data from 2000, 2005, 2011, 2016, and 2019 Ethiopian Demographic Health Surveys (EDHSs). The 2019 updated version of the world health organization (WHO's) Health Equity Assessment Toolkit (HEAT) software was used to analyze the data. Six measure of inequality was calculated: ratio (R), differences (D), relative index of inequality (RII), slope index of inequality (SII), population attributable fraction (PAF) and population attributable risk (PAR). The findings were disaggregated by the five equity stratifiers: economic status, education, place of residence, regions and sex of the child.

### Results

This study showed an erratic distribution of DPT3 immunization services utilization in Ethiopia. The trends in national DPT3 immunization coverage increased from 21% in (2000) to 62% in (2019) (by 41 percentage points). Regarding economic inequality, DPT3 immunization coverages for the poorest quintiles over 20 years were 15.3% (2000), and 47.7% (2019), for the richest quintiles coverage were 43.1 (2000), and 83.4% (2019). However, the service utilization among the poorest groups were increased three fold compared to the richest groups. Regarding educational status, inequality (RII) show decreasing pattern from

**Funding:** The fund is from Addis Ababa University additional support from Professor Helmut

**Competing interests:** All Authors have no any competing interest

7.2% (2000) to 1.5% in(2019). Concerning DPT3 immunization inequality related to sex, (PAR) show that, sex related inequality is zero in 2000, 2005 and in 2019. However, based on the subnational region level, significance difference (PAR) was found in all surveys: 59.7 (2000), 51.1 (2005), 52.2 (2011), 42.5 (2016) and 30.7 (2019). The interesting point of this finding was that, the value of absolute inequality measures (PAR) and (PAF), are shown a decreasing trends from 2000 to 2019, and the gap among the better of regions and poor regions becoming narrowed over the last 20 years. Concerning individual and community level factors, household wealth index, education of the mother, age of respondent, antenatal care, and place of delivery show statically significant with outcome variable. Keeping the other variables constant the odds of an average child in Amhara Region getting DPT3 immunization was 54% less than for a child who lived in Addis Ababa (OR: 0.46, 95% CI: 0.34 – 0.63). Respondents from households with the richest and richer wealth status had 1.21, and 1.26 times higher odds of DPT3 immunization services utilization compared to their counterpart (OR: 1.21, 95% CI: 1.04 -1.41) and (OR: 1.26, 95% CI: 1.13 – 1.40) respectively.

## Conclusion

We conclude that DPT3 immunization coverage shows a growing trend over 20 years in Ethiopia. But inequalities in utilization of DPT3 immunization services among five equality stratifies studied persisted. Reasons for this could be complex and multifactorial and depending on economic, social, maternal education, place of residence, and healthcare context. Therefore, policy has to be structured and be implemented in a ways that address context specific barriers to achieving equality among population sub-groups and regions.

## Introduction

Global health initiatives are gradually giving emphasis to health equality as an important element in attaining universal health coverage [1]. The Global Vaccine Action Plan approved by the 194 Member States of the World Health Assembly in 2012 supports an equity centered strategy to increase vaccine coverage, by giving due emphasize both at national and subnational levels. At the end of 2015, the objective of all countries was to attain 90% and 80% coverage at national and subnational levels respectively, of the DPT3 vaccine as well as all vaccines included in routine immunization programs by 2020 [1, 2]. Hoping that giving vaccine is believed to be the most effective interventions for assuring equitable utilization of healthcare services: it is an important strategy in achieving universal health coverage, and an important indicator to measure progress toward sustainable development goals [2].

DPT3 immunization is taken as one of proxy indicators comprehensive childhood immunization, suggesting that children are receiving all recommended basic vaccines for their age. It also an indication of a country's promotion of routine immunization services [3] and an indicator of healthcare service quality because it assesses the ability to give complete sequence of the three doses of DPT3 vaccine [4]. It is frequently used as a critical indicator to measure sufficient performance of each health facility and the adequate performance of health institutions linked to immunization program, which is essential in monitoring vaccination coverage and comparing vaccination rates of different countries [5].

Globally, in 2022, 14.3 million infants did not receive an initial dose of DTP vaccine, pointing to a lack of access to immunization and other health services, and an additional 6.2 million

are partially vaccinated. Of the 20.5 million, just around 60% of these children live in 10 countries: Angola, Brazil, the Democratic Republic of the Congo, Ethiopia, India, Indonesia, Mozambique, Nigeria, Pakistan and the Philippines [6].

Developed countries have achieved coverage of more than 80 percent with childhood vaccines. While, coverage in the poorest countries has been significantly lower and the poorest countries still have coverage of less than 50 percent [6, 7].

The biggest challenge in achieving high coverage and equality include poor quality district planning; inadequate funding of district staff and operational costs, leading to low quality and unreliable services; and inadequate monitoring and supervision of immunization activities, Political commitment to maintaining vaccination infrastructure in the face of competing priorities. Monitoring data at subnational levels is critical to helping countries prioritize and tailor vaccination strategies and operational plans to address immunization gaps and reach every person with life-saving vaccines [7].

The World Health organization(WHO) recommended the Global Vaccine Action Plan 2011–2020, as a charter for increasing immunization coverage: It recognizes that vaccine has to be universally accessible to every and each individual and the immunization coverage has to reach to the level of 90% [8]. As part of this international initiative, Ethiopia has done a lot struggle to increase its DPT3 immunization coverage high priority. Starting about 25 years ago in Ethiopia, DPT3 immunization coverage has increased as the government implemented a Reaching Every District approach. This strategy aimed at reaching all of the country's districts utilize health extension program enhanced routine immunization activities and community outreach [9]. In spite of all these remarkable efforts, the DPT3 immunization coverage remains low [10].

Furthermore, pocket studies in Ethiopia show that, vaccination coverage vary according to geographical, socioeconomic and demographic factors; hence, inequalities in immunization coverage are found by household economic status, place of residence, mother's education and sex of a child [11]. However, all the studies did not consider trend of DPT3 immunization coverage overtime and did not address inequality in DPT3 vaccination uptake by socioeconomic and regional differences [12, 13].

To understand immunization distribution among population subgroups it is so vital not to relay up on the national average coverage data [14]. Relevant parameters for determining immunization distribution among a country's population subgroups include trends of inequalities based on household wealth status, maternal education and place of residence has to be studied [12]. Evaluating trend of inequality regarding immunization at this subpopulation groups may detect the gaps and suggests new strategies aimed at improving coverage for DPT3 immunization in under vaccinated population subgroups [11, 13, 14].

Therefore, to the best of our knowledge there is lack of studies regarding level and trends of DPT3 immunization inequalities in Ethiopia context. Hence, the current study aimed to investigate level and trends of inequality in DPT3 immunization service utilization over 20 years in all of Ethiopia's regions using data from the five Ethiopian Demographic Health Surveys (EDHSs) carried out between 2000 and 2019.

## Materials and methods

### Study setting and period

Ethiopia is one of the poorest countries in Africa, with a per capita gross national income of $960 [15]. However, it aims to reach lower-middle-income status by 2025 [15]. Over the past 15 years, Ethiopia's economy has been among the fastest growing in the world (at an average of 9.5% per year). The consistently high economic growth over the last decade has resulted in

positive trends in poverty reduction in both urban and rural areas. The share of the population living below the national poverty line decreased from 30% in 2011 to 24% in 2016, and human development indicators improved as well. However, health service related gains are modest when compared to other countries that saw fast growth, and inequalities have increased in recent years. Furthermore, conflicts in various parts of Ethiopia may undermine the economic and social development progress the country has achieved in recent years [16].

Ethiopia has a three-tier health care delivery system. The first lowest level is a woreda/district health system comprising primary hospitals, health centers and their satellite health posts. The first two parts are engaged mostly in curative and preventive health services, while health posts are front-runners in the provision of preventive health services, with a focus on maternal and child health. The second tier is a general hospital; the third tier consists of a specialized teaching and referral hospitals, which entirely focuses on curative healthcare services. Using these tiers system the country provides all recommended vaccinations such as DPT3 immunization services for children age 12–23 months.

### Data sources

The study used data from 2000, (9,543), 2005(8,974), 2011(10,795), 2016(5,980) and 2019 (3,208) (EDHSs). A total of 38,500 mothers who have under five children were involved and all the methods and the data used for this study is comparable across all rounds of the DHSs. The 2019 updated offline version of World Health Organization (WHO's) Health Equity Assessment Toolkit (HEAT) software was used as both the source of data and analytic tool these functions are detailed elsewhere [17, 18].

The Health Equity Monitor (HEM) database is a major component of the Global Health Observatory, the main statistics repository of the World Health Organization (WHO). HEM was created as a resource to promote and enable global and national health inequality monitoring, particularly within low- and middle-income countries, where data availability may be limiting. The practice of health inequality monitoring requires health data that are disaggregated by population Subgroups (i.e. by dimensions of inequality); to this end, HEM contains high-quality, disaggregated health data that are comparable across countries and over time. Currently, reproductive, maternal, newborn and child health is the featured topic of (HEM), which contains indicators categorized under the subthemes of reproductive health interventions, maternal health interventions, and newborn and child health interventions [19]. Technical information about the data, including detailed indicator definitions, is accessible from the data repository or through the theme page.

EDHSs used a two-stage cluster design to draw samples. The territory of Ethiopia was stratified into various strata for the purpose of sampling, and designing enumeration areas (i.e., the primary sampling units or clusters. The samples were selected through Probability Proportional to Size [20] in the first stage from each stratum [21]. In the second stage, 28 to 30 households were recruited from within each primary sampling unit independently. The nationally representative EDHS surveys collect information including topics such as demographic, socioeconomic, maternal health services, and childhood illness. Women aged 15 to 49 years were the major source of data for the surveys, although data were also collected regarding under five children. More details on the methodologies of the respective surveys are presented by Ethiopia's Central Statistics Agency [21, 22].

### Variables and inequality measures

DPT3 immunization services utilization was the primary outcome variables for which inequality was assessed. It was calculated as the percentage of infants aged 12–23 months who received

three doses of the combined diphtheria, tetanus toxoid and pertussis (DPT3) vaccine in a given year to the total number of children aged 12–23 months surveyed.

Five equality stratifies, were selected and used for the data analysis: maternal education, place of residence, economic status, sex of the child and subnational regions of the country. Maternal educational status was categorized as no education, primary education and secondary education and higher. The EDHS uses a wealth index as a proxy indicator for economic status; is constructed using household assets and ownership through principal component analysis (PCA) as described by Shibre et al. [20] and is believed to be comparable across all the survey years. Economic status in this study was categorized as poorest, poor, middle, rich, and richest. Place of residence was categorized as urban or rural. The subnational regions included the nine regions and two city administrations of the country; no sequence was shown in subnational region segregations.

## Data analysis

The 2019 updated version of the WHO's (HEAT) software [21], was used to analyze the socio-demographic, socioeconomic and urban-rural inequalities in Ethiopia over the last 20 years (2000 to 2019). Datasets were investigated and disaggregated by the five equity stratifiers: economic status, maternal education, place of residence, sex and subnational region of the country. These were presented using the 6 summary measures of health inequality chosen from the pool of 15 measures available in the software: difference (D), ratio (R), population attributable fraction (PAF), population attributable Risk (PAR), slope index of inequality (SII) and relative index of inequality (RII) [17, 21]. These measures were chosen because they have broader use in healthcare inequity research applications [18, 22]. In equality studies, both simple and complex summary measures are calculated for each equality stratifies as recommended by WHO. Briefly, (D) and (R) are simple measures of health inequality, whereas (SII), (RII), (PAF), and (PAR) are complex measures. Also, whereas difference (D), (SII), and (PAR), are absolute measures, (RII), (R) and (PAF) are relative measures.

Simple measures work for pairwise comparison of a health indicator, but they do not consider subpopulations for an indicator with more than two categories (e.g., wealth index, subnational region). Complex measure estimates consider all the categories within an indicator. To show inequality, the lower and upper bounds of the (CI) do not include zero (0) for (D) and (SII). For (R) and (RII), inequality exists if (CI) does not include one (1). To assess inequality across years, (CI) of the summary measures for each survey year should not overlap. Inequities in Maternal Health Services (MHS) utilization were measured by Horizontal Inequity Indices (HII) and concentration curve following Wagstaff method of analysis [23].

To identify determinant factors for inequality in utilization of DPT3 services, R Software was used in analyzing the data. The data was analyzed in to two hierarchal levels. The variables were categorized in to community level and individual level factors: under community level, surveys year, region and place of residence were categorized, whereas, wealth index, maternal education, age of the mother, sex of the child, birth order, antenatal care (ANC), and place of delivery of the child were categorized under individual level factors. Multilevel Binary logistic regression for the two ordered measures with ranking of weighted samples from most disadvantaged (poorest and uneducated - ranked 0) to most advantaged (secondary education or higher and the richest subgroups - ranked 1). For region low or high coverage/ urban rural difference was used; (95%) CI was computed around point estimates to measure statistical significance. Finally the findings were presented using texts, tables and figures for each of the five Ethiopia Demographic and Health Surveys (EDHSs) periods. The disaggregation included the computed point estimates with a corresponding 95% confidence Interval (CI).

**Operational definitions.** **Difference (D)** = Measures the difference in health between two population subgroups and measures absolute inequality. $D = yhigh - ylow$.

**Ratio (R)** = R is a relative measure of inequality that shows the ratio of two population subgroups. It is calculated for all inequality dimensions, provided that subgroup estimates are available for the two subgroups used in the calculation of R. Calculation R is calculated as the ratio of two subgroups: $R = yhigh/ylow$.

**Slope index of inequality (SII)** = represents the difference in estimated values of a health indicator between the most-advantaged and most-disadvantaged (or vice versa for adverse health indicators), while taking into consideration all the other subgroups. SII is an absolute measure of inequality that takes into account all population subgroups. It is calculated for ordered dimensions with more than two subgroups, such as economic status. For favorable health indicators, the difference between the estimated values at rank 1 ($v1$) and rank 0 ($v0$) (covering the entire distribution) generates the SII value: $SII = v1 - v0$.

**Relative index of inequality (RII)** = represents the ratio of estimated values of a health indicator of the most-advantaged to the most disadvantaged (or vice versa for adverse health indicators), while taking into account all the other subgroups. RII is a relative measure of inequality that takes into account all population subgroups. It is calculated for ordered dimensions with more than two subgroups, such as economic status. For favorable health indicators, the ratio of the estimated values at rank 1 ($v1$) to rank 0 ($v0$) (covering the entire distribution) generates the RII value: $RII = v1/v0$.

**Population attributable risk (PAR)** = PAR shows the potential for improvement in setting (national) average that could be achieved if all population subgroups had the same level of health as a reference group. PAR is an absolute measure of inequality that takes into account all population subgroups. Calculation PAR is calculated as the difference between the estimate for the reference subgroup $yref$ and the setting (national) average μ: $PAR = yref - μ$.

**Population attributable fraction (PAF)** = PAF shows the potential for improvement in setting (national) average of a health indicator, in relative terms, that could be achieved if all population subgroups had the same level of health as a reference group. PAF is a relative measure of inequality that takes into account all population subgroups. Calculation PAF is calculated by dividing the population attributable risk (PAR) by the setting (national) average μ and multiplying the fraction by 100: $PAF = PAR/ μ * 100$.

**The concentration index** is defined as twice the area between the concentration curve and the line of equality (the 45-degree line). So, in the case in which there is no socioeconomic-related inequality, the concentration index is zero. Mathematically

$$CI = 2COV(Yi, Ri)/μ,$$

where y is the health variable whose inequality is being measured, μ is its mean, $Ri$ is the ith individual's fractional rank in the socioeconomic distribution (e.g. the person's rank in the income distribution), and cov is the covariance.

**The Gini index** is a measure of the distribution of income across a population. The Gini index can be calculated as the ratio of the area between the perfect equality line and the Lorenz curve (A) divided by the total area under the perfect equality line (A + B).

$$GI = A/(A + B)$$

The result of this analysis is shown in graph by Lorenz curve (Fig 7).

### Ethical considerations

The study used data freely accessible from the five (EDHSs). All EDHS surveys were approved by ICF and the Institutional Review Board of respective country to ensure that the protocol of all EDHS surveys is followed the U.S. Ethical approval for the study was obtained from the Institutional Review Board of the College of Health Sciences at Addis Ababa University.

## Results

### Trends of DPT3 immunization coverage

The trend in DPT3 immunization services utilization was assessed across the five equity stratifies over 20 years from Ethiopian demographic surveys (2000–2019) data. The point estimate of the proportion of DPT3 immunization over 20 years was computed (Table 1). Accordingly the national DPT3 immunization coverage increased from 21% in (2000) to 33% in (2005), then 37% in (2011), 53% in (2016), and sharply increased to 62% in (2019).

### Levels and trends of in DPT3 immunization services utilization based on inequality dimensions

This study revealed that, DPT3 immunization was varied based on educational status, economic status, place of residence, and subnational regions with in the country. There was more high coverage for advantaged groups compared to disadvantaged groups. Regarding economic status, DPT3 immunization coverage for the poorest quintiles over 20 years had been 15.3% (2000), 25.6% (2005), 26% (2011), 36.4% (2016), and 47.7% (2019) for the richest quintiles it was 43.1 (2000), 49.2% (2005), 63.6% (2011), 76.3% (2016), and 83.4% (2019) (Table 2) and (Fig 1). This indicates statistically significance differences in coverage between the poorest and richest sub-groups. Although the higher wealth index groups shows higher coverage in DPT3 service utilization compared to the poorest group, the utilization is increased three fold among the poorest groups during the mentioned periods, which was 15.3%(CI:11.7, 19.7) in 2000 and 47.7%(CI: 36.6, 59.0) in 2019, which is much higher than the higher quintile groups. Regarding maternal education levels, DPT3 immunization coverage was higher among secondary school sub-groups in all the five surveys compared to no education sub-groups (Table 2).

Concerning residence, the current study revealed urban–rural differences in the DPT3 coverage; higher coverage was found in urban sub-groups, especially in the 2000 and 2005 surveys. For instance, a 34 percent point difference in (2000) and a 37 percentage point's difference in 2005. Surprisingly, for rural residence coverage increased consistently over the five survey periods. Whereas for urban residence coverage increased from 2000 to 2005, dropped in 2011 rose again in 2016 and dropped again in2019 (Table 2). Regarding sex- based distribution of DPT3 immunization services utilization, the coverage shows insignificant disparities between

**Table 1. Samples summary and trend of DPT3 immunization coverage in Ethiopia from 2000–2019.**

| EDHs, Year | Number of children participated in each EDHs survey at national level | Sample taken from each EDHS for DPT3 immunization based on inclusion criteria | DPT3 Immunization Coverage per Year |
|---|---|---|---|
| 2000 | 15367 | 9543 | 21% |
| 2005 | 14070 | 8974 | 33% |
| 2011 | 16515 | 10795 | 37% |
| 2016 | 15683 | 5980 | 53% |
| 2019 | 885 | 3208 | 62% |
| Total | **70,520** | **38,500** | |

**Table 2. Levels and trends of in DPT3 immunization services utilization based on inequality measures: Evidence from Ethiopian demographic and health surveys, 2000–2019.**

| Dimensions of inequality | 2000 | | 2005 | | 2011 | | 2016 | | 2019 | |
|---|---|---|---|---|---|---|---|---|---|---|
| | %(95%)CI | Pop | %(95%)CI | Pop | %(95%)CI | Pop | %(95%)CI | Pop | %(95%)CI | Pop |
| **Economic status** | | | | | | | | | | |
| Quintle1(poorest) | 15.3(11.7,19.7) | 21.3 | 25.6(19.33.6) | 24 | 26(20.2,32.9) | 22.9 | 36.4(29.5,44) | 25.2 | 47.7(36.6,59) | 21.1 |
| Quintile 2 | 11.9(8.2,16.9) | 20.7 | 27.3(21.7,33.8) | 21.2 | 29.4(23,36.6) | 21.7 | 50.4(42.6,58.2) | 19.8 | 59.5(46.3,71.4) | 20.8 |
| Quintile 3 | 15.4(11.8,19.9) | 21.6 | 33.7(27.5,40.7) | 20.3 | 31.4(24.4,39.4) | 20.4 | 51.4(43.5,59.2) | 22.4 | 52.1(41.3,62.8) | 17.4 |
| Quintile 4 | 24.9(18.9,32.2) | 19.5 | 32.5(25.2,40.9) | 18.4 | 42.8(35.2,50.8) | 19.1 | 63.2(55.6,70.2) | 18.3 | 63.7(48.8,76.4) | 15.5 |
| Quintile 5 | 43.1(34.4,52.2) | 16.6 | 49.2(41.8,71.8) | 16.1 | 63.6(54.5,71.8) | 15.9 | 76.3(65.5,84.5) | 14.4 | 83.4(58,94.8) | 25.3 |
| **Education** | | | | | | | | | | |
| No Education | **16.3(13.3,19.2)** | 79.6 | 28.5(24.5,32.9) | 77.6 | 31.7(27.5,36.2) | 67.7 | 45.3(40.6,50.2) | 62.7 | 55.8(49.4,62.1) | 45.2 |
| Primary School | **35(28.5,42.1)** | 14.9 | 42(34.3,50) | 17.5 | 43.5(37.5,49.7) | 27 | 62.3(55.7,68.5) | 28.8 | 65.1(54.7,74.2) | 40.6 |
| Secondary school + | **53.9(38.7,68.4)** | 5.5 | 65.6(54.575.3) | 4.9 | 72.8(59.5,83.0) | 5.3 | 79.8(65.6,89.1) | 8.5 | 75.6(57.5,87.7) | 14.2 |
| **Residence** | | | | | | | | | | |
| Rural | **17.6(15.2,20.3)** | 89.5 | 29.8(26.1,33.8) | 92.1 | 32.9(29,37.1) | 85.8 | 49.7(45.3, 54.2) | 88.4 | 56.1(49.6,62.3) | 69.5 |
| urban | **51.4(38.8,63.7)** | 10.5 | 66.8(57.3,75.1) | 7.9 | 62(52,71) | 14.2 | 79.5(66.1, 88.5) | 11.6 | 76.9(57.4,89.1) | 30.5 |
| **Sex** | | | | | | | | | | |
| Female | **19.4(16.1,23.2)** | 48.3 | 30.4(26,35.2) | 48.9 | 39.3(34.4,44.4) | 47.7 | 53.3(48.1,58.5) | 53.8 | 60.1(51.6,68) | 51.8 |
| Male | **22.8(19.4,26.5)** | 51.7 | 34.9(30.3,39.7) | 51.1 | 35.(30.4,39.9) | 52.3 | 52.9(47.6,58.2) | 46.2 | 64.9(57.4,71.8) | 48.2 |
| **Subnational Region** | | | | | | | | | | |
| **Addis Ababa** | 80.9(66.6,89.9) | 1.5 | 83.8(71,91.6) | 1.7 | 89.2(78.8,94.8) | 2.2 | 95.7(88.9,98) | 2.6 | 93.1(84.3,97.2) | 3.3 |
| **Afar** | 1.1).1,7.3) | 0.9 | 4.6(1.8,11.2) | 1.0 | 11.6(6.8,18.9) | 0.9 | 20.1(11.6,32.5) | 1.0 | 27(17.4(39.3) | 1.5 |
| **Amhara** | 20.6(15.3,27.2) | 26,3 | 32.1(25.4,39.7) | 25.7 | 39.4(31.8,47.7) | 23.1 | 63.8(53.5,72.9) | 18.2 | 82.2(70.8,89.9) | 21.2 |
| **Benishangul Gumuz** | 16.7(10.2,26.1) | 0.9 | 30.7(19.2,45.1) | 0.9 | 42.9(32.7,53.6) | 1.2 | 76.2(63.8,85.3 | 1.0 | 80.5(70.7,87.7) | 1 |
| **Dire Dawa** | 52.4(40.8,63.8) | 0.3 | 62.5(50.4,73.3) | 0.4 | 76.1(67.5,82.9) | 0.4 | 84.9(75.3,91.2) | 0.5 | 74.5(62.7,83.6) | 0.6 |
| **Gambella** | 12.7(6.6,23.6) | 0.2 | 20.3(11.9,32.3) | 0.3 | 29.4(20.7,39.9) | 0.4 | 54.8(42.1,67) | 0.3 | 69(54.6,80.4) | 0.4 |
| **Harari** | 50.7(42.4,58.9) | 0.2 | 45.8(36.7,55.2) | 0.2 | 54.4(42.7,65.6) | 0.3 | 58.7(47.2,69.3) | 0.2 | 54.9(41.4,67.8) | 0.2 |
| **Oromia** | 16.6(12.9,21.2) | 42.2 | 28.8(22.1,36.6) | 36.8 | 27.1(21.5,33.6) | 42 | 39.9(33.1,47.1) | 44 | 53.6(40.9,65.8) | 39.4 |
| **SNNPR** | 16.9(13.2,21.5) | 20.6 | 35.6(28.8,43) | 21.7 | 38.5(30.8,46.8) | 20.2 | 59(50,7,66.9) | 20.9 | 56.3(41.6,70) | 19.4 |
| **Somali** | 24.4(12.3,42.8) | 1.1 | 5.6(1.4,20.2) | 4.2 | 25.8(15.9,39) | 2.6 | 36.3(26.6,47.2) | 3.8 | 26.2(15.1,41.5) | 5.4 |
| **Tigray** | 56.8(47.8,65.5) | 5.5 | 52.1(44.3,59.7) | 7.2 | 74.3(66.1,81) | 6.7 | 81.4(71.1,88.2) | | 84.4(72.3,91.8) | 7.5 |

male and female children in all surveys (Table 2). One very important finding from this study is a huge variation existed in DPT3 immunization coverage across subnational regions from 2000 to 2019. Except in Somalia, Harari and Tigray coverage increased sharply from 2000 to 2005 in all regions. In the latest survey (2019) the coverage was lowest in Somali, Southern nations, Nationalities and people (SNNP), Harari and Dire Dawa. But in the second and the third surveys coverage increased sharply in almost all regions, decreasing only in Oromia. Continuous incremental coverage was observed in Amhara, Afar, Benishangule Gumuz and Gambella regions throughout entire study period (Table 2) and (Fig 2).

## Trends of inequality in DPT3 immunization services utilization based on summary measure of inequality

Both absolute (D, SII, PAR) and relative (R, RII, PAF) measures of inequality were taken in this study (Table 3). The finding of all this summary measure of inequality show pro-poor inequality in DPT3 immunization in Ethiopia across the five surveys from 2000 to 2019. The

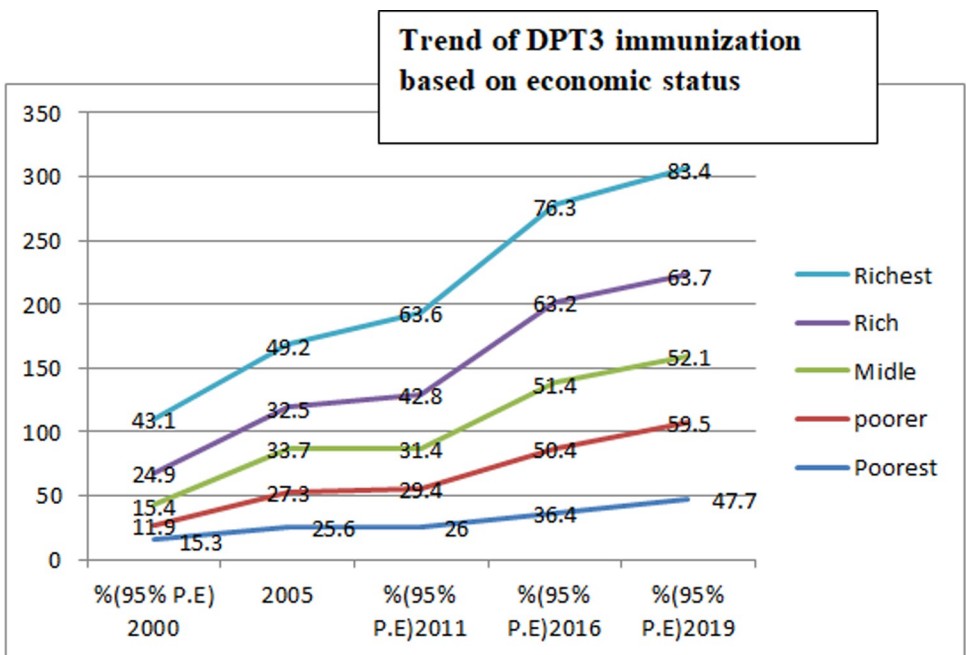

**Fig 1. Trend and coverage of DPT3 immunization based on economic status in Ethiopia from 2000 to 2019.**

absolute measures of inequality, reviled wealth related inequality existed in 2000, 2011, 2016 and 2019. Wealth index (R) inequality consistently reduced from 2000 to 20019 throughout the surveys in DPT3 immunization except in 2011. However, the (SII) measure shows 32.4% in (2000) and 41.9% in (2019) revealing high pro-rich inequalities exist in DPT3 immunization based on wealth index. The wealth related an absolute complex measure (PAR) was 43% in (2000) and 16.7 in (2019) indicating a decreasing pattern in DPT3 immunization inequality in the last 20 years. The education related absolute measure (D) shows a leveled off over the course of the surveys, except for a sharp increase in 2011. Relative inequality measures (RII) for education show decreasing pattern from 7.2% in (2000) to 1.5% in (2019), this indicate educational related inequality between advantaged and disadvantaged group is becomes narrowed (Fig 3). As with education status, place of residence show unequal distribution of DPT3 immunization coverage. Absolute measures of inequality (D) and (R) revile continuous pattern of decreasing in residence based inequality over the surveys (Table 3) and (Figs 4 and 5). Regarding DPT3 immunization inequality related to sex, (PAR) show sex related inequality is zero in 2000, 2005. and in 2019 indicating equality in DPT3 service utilization by sex population sub-groups (Fig 6). However, regarding subnational region, significance difference (PAR) was found in all surveys; 59.7 in (2000), 51.1 in (2005), 52.2 in (2011), 42.5 in (2016), and 30.7 in (2019). Relative inequality measure (PAF), was 282.4 in (2000) and 49.2 in (2019. The interesting point from this finding was the value of (PAR) and (PAF) were decreasing from (2000 to 2019), this show that DPT3 immunization inequality among subnational regions were decreased from (2000) to (2019) over the last 20 years and equality in services utilization became improved (Table 3).

The concentration curve for this study show that, DPT3 immunization utilization distribution is more concentrated in high wealth index population groups (Fig 7A–7D).

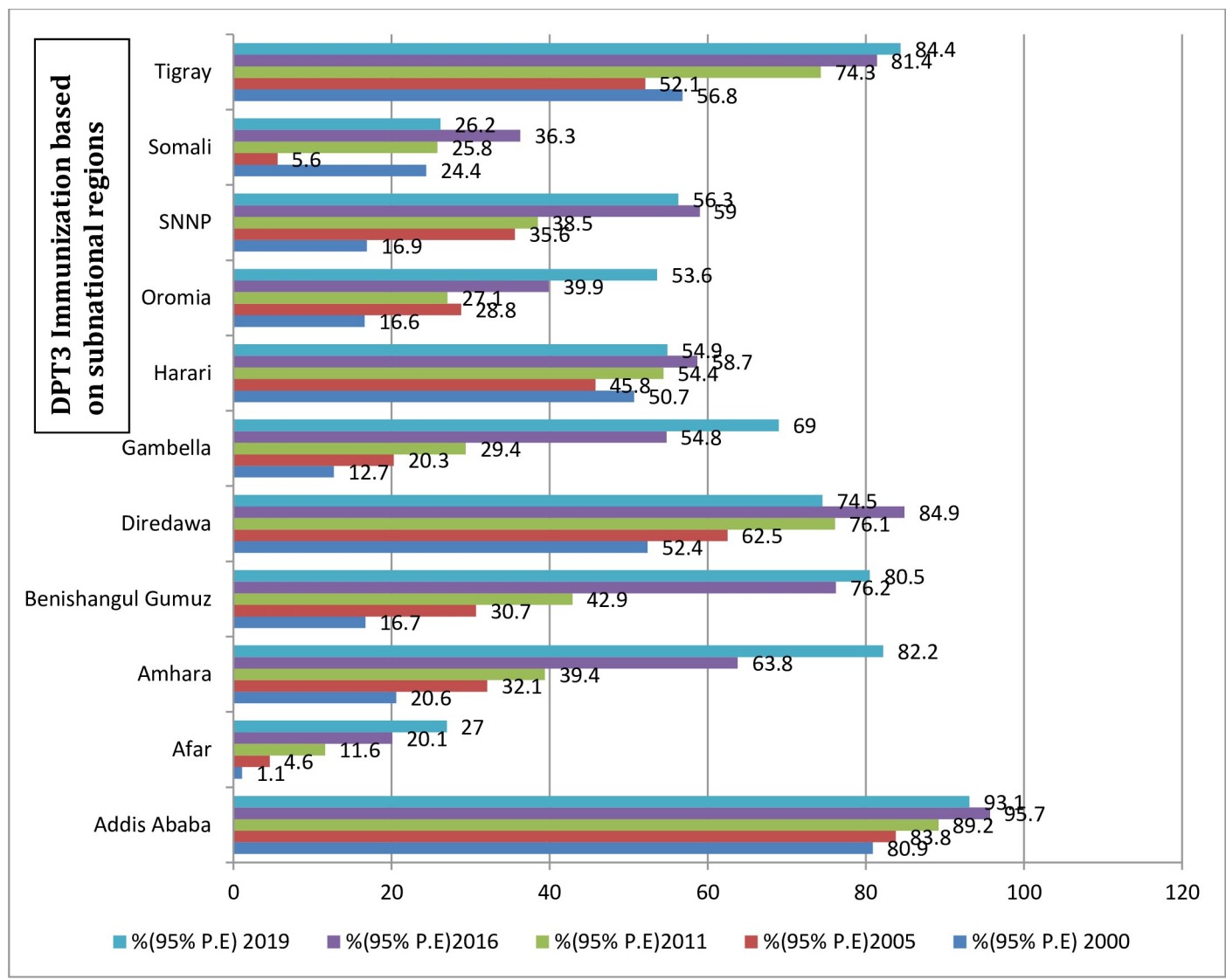

**Fig 2. Trends and coverage of DPT3 immunization based on subnational regions in Ethiopia from 2000 to 2019.**

## Multilevel analysis of DPT3 immunization utilization

Multilevel Binary logistic regression was carried out to identify determinant factors. From community level factors, year, region and residence were statistically significant with DPT3 immunization. Regarding individual level factors, household wealth index, education of the mother, sex, age of respondent, antenatal care utilization, place of delivery, birth order showed statically significant with DPT3 immunization service utilization. After controlling other variables, children in the 2005 EDHS have 1.48 (95% CI: 1.26 – 1.75) times higher odds of getting DPT3 immunization than those in the 2000 EDHS. In another way, keeping the other variables constant the odds of getting DPT3 immunization for a child who lived in Amhara region was 54% less than for a child who lived in Addis Ababa(OR: 0.46, 95% CI: 0.34 – 0.63). Respondents from households with the richest and richer wealth status had 1.21, and 1.26 times higher odds of getting DPT3 immunization service utilization compared to their counterpart (OR: 1.21, 95% CI: 1.04 – 1.41) and (OR: 1.26, 95% CI: 1.13 – 1.40) respectively. Moreover, each

**Table 3. Trends of DPT3 immunization services utilization based on summary measures of inequality from EDHS (2000–2019).**

| Dimension of inequality | Measure of inequality | Year | | | | |
|---|---|---|---|---|---|---|
| | | 2000% (95% CI) | 2005% (95% CI) | 2011% (95% CI) | 2016% (95% CI) | 2019% (95% CI) |
| Economic status | | | | | | |
| | D | 47.5(28,67) | 47.6 (35.7,59.5) | 54(40.9,67.2) | 54.2(39.6,68.8) | 42.9(15.1,70.7) |
| | PAF | 203.4(181.4,225.4) | 119.6 (103,136.) | 110.9 (96.3,125.5) | 61.5(50.9,72.1) | 26.8(13.2,40.4) |
| | PAR | 43(38.4,47.7) | 39.1(33.7,44.6) | 41.1(35.7,46.5) | 32.7(27,38.3) | 16.7(8.3,25.2) |
| | R | 3.8(2.4,6.1) | 3(2.1,4.2) | 3.2(2.3,4.6) | 2.7(1.9,3.9) | 2.2(1.3,3.5) |
| | RII | 5.1(3.7,6.9) | 2.2(1.8,2.8) | 3.1(2.5,3.9) | 2.4(2, 2.8) | 2.1(1.7,2.5) |
| | SII | 32.4(26.6, 38.3) | 25.7(18.6,32.7) | 39.7(33.146.4) | 43(36.4,49.6) | 41.9(32.9,51) |
| Education | | | | | | |
| | D | 37.6(22.2,53.1) | 37.1(25.8,48.4) | 41.1(28.5,53.5) | 34.4(21.8,47.1) | 19.3(3.2,36.6) |
| | PAF | 154.9(149.9,159.8) | 100.7(96.9,104.5) | 96.5(92.4,100.7) | 50(46.8,53.3) | 21.1(15.8,26.4) |
| | PAR | 32.7(31.7,33.8) | 32.9(31.7,34.2) | 35.8(34.2,37.3) | 26.6(24.9,28.3) | 13.2(9.9,16.5) |
| | R | 3.3(2.4,4.6) | 2.3(1.9,2.9) | 2.3(1.9,2.8) | 1.8(1.5,2.1) | 1.4(1.1,1.7) |
| | RII | 7.2(5.1,10.1) | 3.1(2.3,4.2) | 2.6(2,3.3) | 2.4(2,2.8) | 1.5(1.3,1.8) |
| | SII | 40.5(33.6,47.4) | 35.8(27.2,44.4) | 34.1(26,42.1) | 42.6(34.9,50.2) | 25.8(15.1,36.5) |
| Residence | | | | | | |
| | D | 33.8 (20.9,46.7) | 37 (27.2,46.7) | 29 (18.6,39.4) | 29.7(17.7,41.8) | 20.8(3.7,38) |
| | PAF | 143 (139.2,146.8) | 104.2 (101.9,106.5) | 67.3 (64.6,69.9) | 49.5(48,51) | 23.2(20.1,26.3) |
| | PAR | 30.2(29.4,31) | 34.1(33.1,34.8) | 24.9(23.9, 25.9) | 26.3(25.5,27.1) | 14.5(12.5,16.4) |
| | R | 2.9(2.2,3.9) | 2.2(1.9,2.7) | 1.9(1.5, 2.3) | 1.6(1.4,1.9) | 1.4(1.1,1.7) |
| | RII | NA | NA | NA | NA | NA |
| | SII | NA | NA | NA | NA | NA |
| Sex | | | | | | |
| | D | -3.4(-8.4,1.6) | -4.5(-11,2.1) | 4.3(-2.6,11.2) | 0.4(-7,7.9) | -4.9(-15.8,6.1) |
| | PAF | 0(-7.9,7.9) | 0(-6.3,6.3) | 6.1(0.6,11.7) | 0.4(-4.1,4.8) | 0(-4.9,4.9) |
| | PAR | 0(-1.7,1.7) | 0.(2.1,2.1) | 2.3(0.2,4.3) | 0.2(-2.2,2.6) | 0(-3.1,3.1) |
| | R | 0.9(0.7,1.1) | 0.9(0.7,1.1) | 1.1(0.9,1.4) | 1(0.9,1.2) | 0.9(0.8,1.1) |
| | RII | NA | NA | NA | NA | NA |
| | SII | NA | NA | NA | NA | NA |
| Subnational | | | | | | |
| | D | 79.8(68.1,91.5 | 79.1(68.2,90.1) | 77.6(68,87.3) | 75.6(64.4,86.9) | 66.9(52.4,81.5) |
| | PAF | 282.4(75.5,489.3) | 156.2,99.4) | 140.8(88.9,192.8) | 80(42.8,117.2) | 49.2(30.5,68) |
| | PAR | 59.7(16,103.5) | 51.1(32.5,69.9) | 52.2(32.9,71.4) | 42.5(22.8,62.3) | 30.7(19,42.4) |
| | R | 76.9(10.7,554.1) | 18.1(7.2,45.1) | 7.7(4.6,12.9) | 4.8(2.8,8) | 3.6(2.1,5.9) |

additional year in respondent's age increased their odds of getting DPT3 immunization by 2% (OR: 1.02, 95% CI: 1.02 – 1.03). Furthermore, mothers who gave birth at health facility (OR: 1.18, 95% CI: 1.07 – 1.31) more likely to utilize DPT3 immunization services than their counterparts (S1 Table).

## Discussion

This study aimed at showing the trends of DPT3 immunization services utilization in Ethiopia that might reveal inequalities in immunization coverage. The study used the WHO HEM database to examine trends over 20 years using data from EDHS surveys from 2000, 2005, 2011, 2016, and 2019.

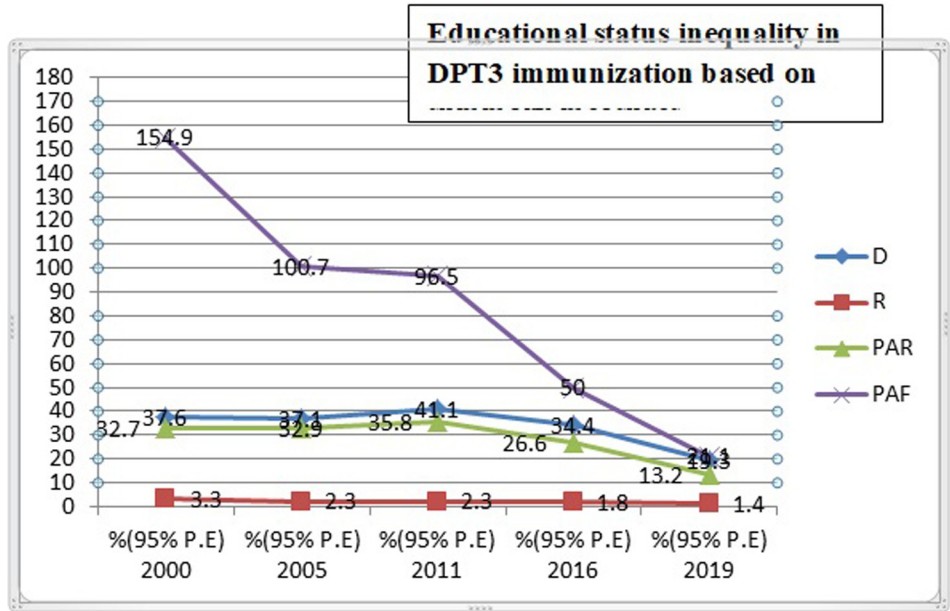

**Fig 3. Trend of educational status inequality in DPT3 immunization based on summary measures in Ethiopia from 2000 to 2019.**

An increasing trend of DPT3 immunization coverage was observed from (21%) 2000 to (33%) 2005, (37%) 2011, (53%), 2016 and to (62%) 2019. Despite the rise in coverage, significant inequalities were found by maternal education, family/household wealth index, place of

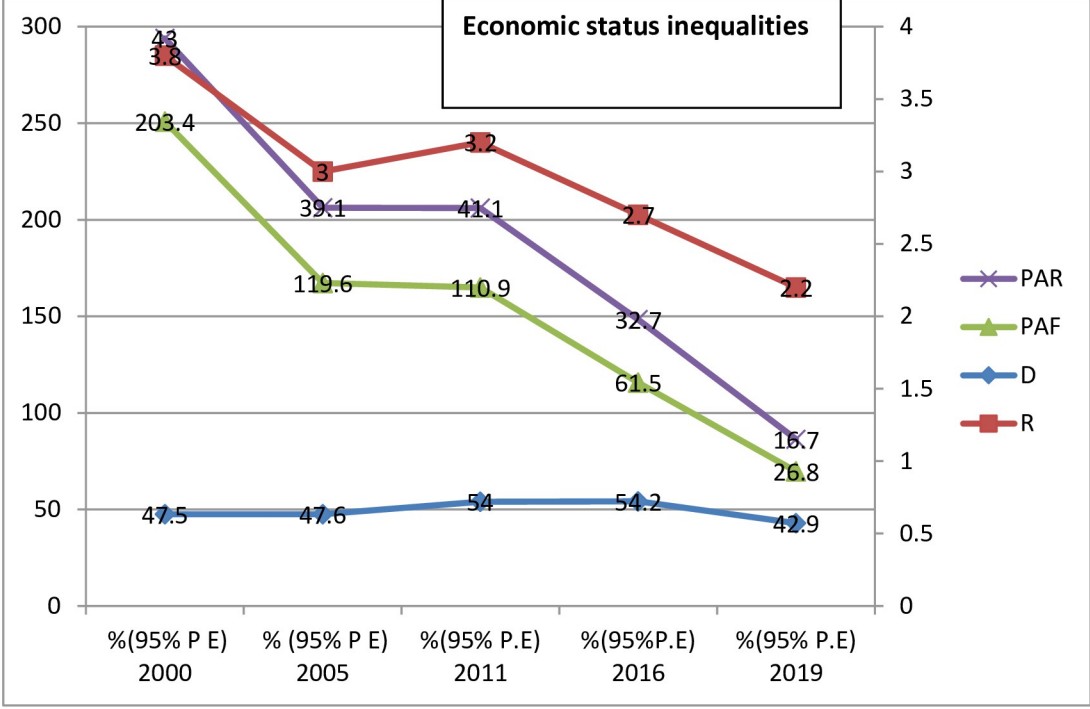

**Fig 4. Trend of economic status inequality in DPT3 immunization based on summary measures in Ethiopia from 2000 to 2019.**

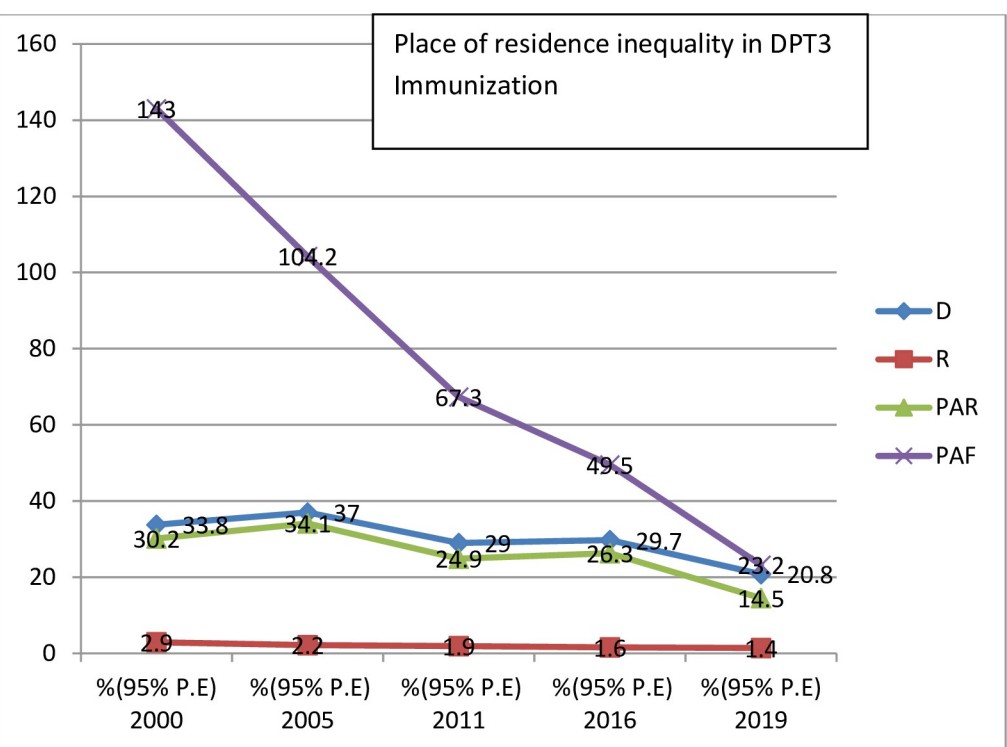

**Fig 5. Trend of residence inequality in DPT3 immunization based on summary measures in Ethiopia from 2000 to 2019.**

residence, and sex of the child. This finding of inequalities that favored the most advantaged households was similar to findings in other studies in Ethiopia [24–26]. From this study, a prominent level of economic inequality was noticed in utilization of DPT3 immunization services among all economic status sub-groups. For instance, DPT3 immunization coverage for the poorest quintiles increased from 15.3% to 47.7% over the 20 years studied (32.4 percentage points) whereas for the richest quintiles the increase for the same period was from 43.1 to 83.4% (40.3 percentage points). The almost 9 percentage point difference indicates a pronounced disparity in utilization of DPT3 immunization services. Similar findings were reported by other studies in Ethiopia and elsewhere in Africa [25, 26]. This could be due to having high income is linked to improved decision making power of the women's, for themselves and for their child as well as lowered fertility rate and increased uptake of preventive health services such as immunization services.

In this study, even though, the higher wealth index groups shows higher coverage in DPT3 service utilization compared to the poorest group, the utilization is increased three fold among the poorest groups during the mentioned periods, which was 15.3%(CI: 11.7, 19.7) in 2000 and 47.7%(CI: 36.6, 59.0) in 2019, this show a significant incremental in the level of service utilization among poor people compared to the richest groups. This might be due to Ethiopian policy gave due emphasis in addressing healthcare service utilization among the poor people; immunization is given free of charge in Ethiopia. Moreover, non-Governmental organization also gives focus on disadvantaged and low socioeconomic groups in covering transportation cost and so on. In turn this may increase service utilization among the poorest groups compared to richest subgroups.

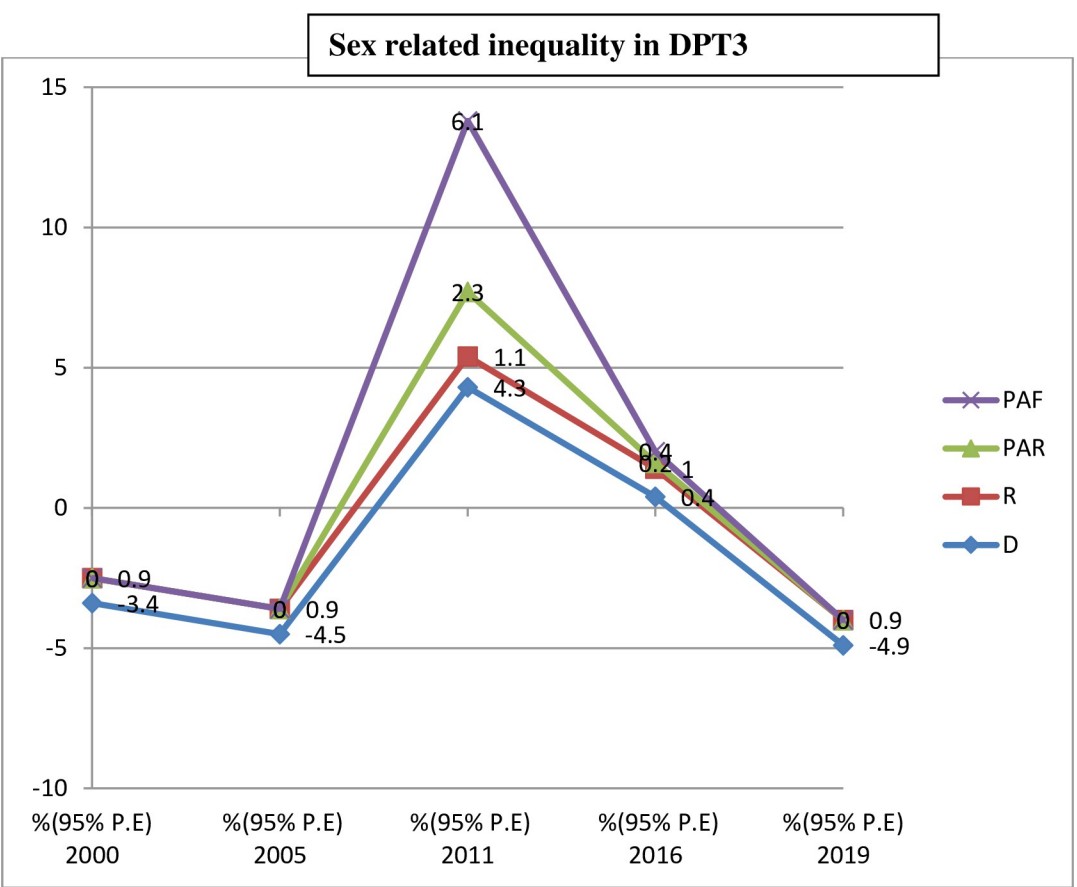

**Fig 6. Trend of sex related inequality in DPT3 immunization based on summary measures in Ethiopia from 2000 to 2019.**

DPT3 immunization coverage increased over the 20-year period for all five economic sub-groups (Table 2). Based on the summary measure of inequality, pro-poor inequality in DPT3 immunization in Ethiopia existed in all five surveys. The wealth index relative inequality (D) consistently declined from one survey to the next except in 2011 and 2016. However, the absolute inequality measure SII was 32.4% in 2000 and 41.9% in 2019, revealing incremental pro-poor inequalities in DPT3 immunization coverage based on economic status. Similarly, the absolute complex measure PAR declined from 43% in 2000 to 16.7 in 2019, indicating the potential for improvement if the average population sub-groups had the same level of health as the reference groups. The reason may be that better economic status may increase mothers' healthcare-seeking behavior and access to health information. Moreover, health literacy was believed to improve communication skills among mothers with higher education, leading to increased odds of childhood immunization services utilization.

This study showed variation in DPT3 immunization services utilization by education status of mothers. DPT3 immunization coverage was statistically significantly higher among mothers with secondary school education than those with primary education and with no education. Several studies have reported an inverse relationship between maternal education and incomplete DPT3 immunization in sub-Saharan Africa, including Ethiopia [26–28, 32, 33]. The reason may be that educated mothers are more likely to seek and understand health-related information and use healthcare and immunization services than uneducated mothers. In addition, education influences maternal attitudes and customary beliefs; increases autonomy in

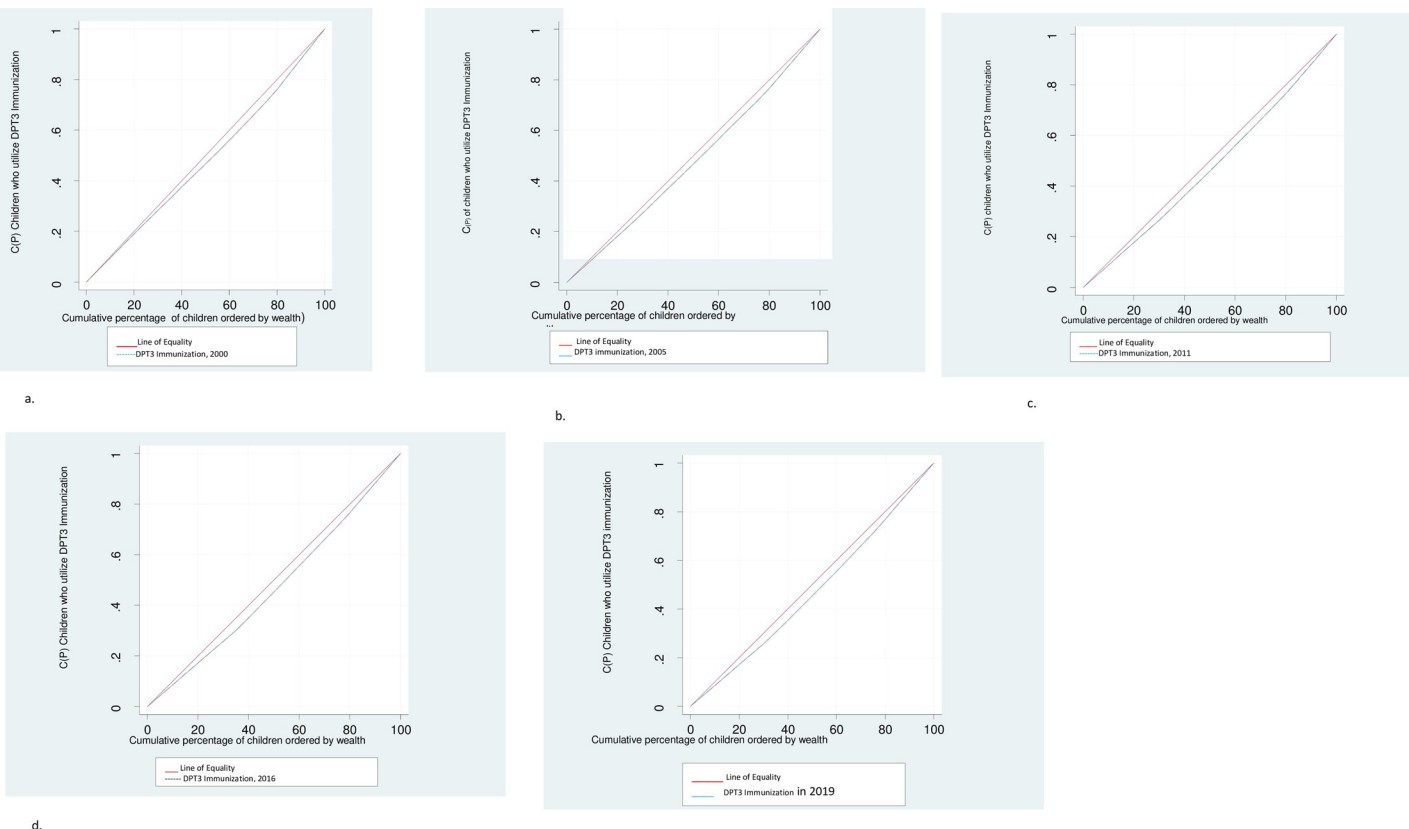

**Fig 7.** a. Concentration curve of DPT3 Immunization EDHS 2000. b. Concentration curve of DPT3 Immunization EDHS 2005. c. Concentration curve of DPT3 Immunization EDHS 2011. d. Concentration curve of DPT3 Immunization EDHS 2016. e. Concentration curve of DPT3 Immunization EDHS 2019.

decision-making; and encourage actions that promote family health provision, including obtaining child immunizations. The education-related absolute measure D decreased from 2000 to 2005, increased sharply in 2011, and decreased again in 2019, meaning it was not constant throughout the study period. Relative inequality measure RII for education impacts declined from 7.2% in 2000 to 1.5% in 2019, indicating that education-related inequality among advantaged and disadvantage groups decreased over the course of the surveys.

Concerning place of residence, DPT3 immunization coverage was 34% higher in urban sub-groups in 2000, 37% higher in 2005, and 21% higher in 2019. Coverage increased 38.5% in rural areas and 25.5% in urban areas during the 20-year study period. Absolute measures of inequality D and R revealed a similar pattern of residence-based inequality over the surveys, corroborating findings from other surveys in Ethiopia [28, 29]. These inequities may be due to distance between places of residence and health facilities and scarcity of child and maternal services in rural areas. Rural women may need to travel longer distances to access immunization services than urban women because health facilities are more concentrated in urban areas than rural areas. One study reported that around 71% of women in Ethiopia encountered at least one type of barrier to accessing healthcare services for themselves and their children (e.g., lack of money, distance to a health facility, and getting permission from the husband to go to a health facility alone) [30, 31].

Regarding sex-based distribution of DPT3 immunization services utilization, slight variations were observed between male and female sub-groups. But regarding summary measures

PAR and PAF, no sex-related disparity was observed. Several previous studies also found no sex-based disparities in immunization services utilization [31–33]. This shows that disparity in health services utilization between male and female genders are becoming narrowed.

One important finding of this study is that DPT3 immunization coverage varied considerably among subnational regions from 2000 to 2019. Except in Somalia, Harari, and Tigray, coverage increased sharply from 2000 to 2005 in all regions. Coverage consistently increased in Amhara, Afar, Benishangul Gumuz, and Gambella regions throughout the study period but declined in 2005 2011 in Oromia (Table 1). These findings corroborate those of other studies in Ethiopia [29, 33]. Based on summary measures, significant differences in PAR values among the subnational regions persisted in all surveys: 59.7 (2000), 51.1 (2005), 52.2 (2011), 42.5 (2016), and 30.7 (2019). The decreasing PAR value indicates that DPT3 immunization inequality among sub-groups decreased during the period. The relative inequality measure PAF also declined, from 282.4 in 2000 to 49.2 2019. In spite of the decrease in relative inequality over time, further improvements in DPT3 coverage are required to narrow the gap between the better-off and poorer regions.

Possible reasons for disparities in DPT3 immunization services utilization in Ethiopia may be differences in healthcare access among regions, long distances to health facilities, lack of transportation, and differences in economic status among populations of these regions. In multilevel analysis, region and place of residence were statistically significantly associated with DPT3 immunization. Individual level factors, including household wealth index, education, age and sex of the mothers, anti natal care utilization, place of delivery, birth order show statically significant with DPT3 immunization service utilization. Similar finding were reporting from Ethiopia and elsewhere in Africa [26, 27, 29, 34–36].

## Strengths and weaknesses

Strength; this finding is based on nationally representative EDHS data, which makes the results generalizable and applicable to nationwide interventions and policy options.

The data resource is one of the largest repositories of disaggregated data from low- and middle-income countries particularly from Ethiopia on areas of child health.

Limitations: of the HEM include the scope of data represented, in terms of the health topic. In some cases, estimates based on low sample sizes pose a limitation.

Disaggregation of the sample population (E.g in to educational status) inevitably decreases the sample size of the point estimate.

## Conclusions and recommendation

Although analysis using six inequality measures showed that DPT3 immunization coverage increased between 2000 and 2019, substantial inequalities remain. A significant inequality in DPT3 immunization was observed in subgroups based on educational status, economic status, place of residence and regions of the country. However, in this study, no sex related inequalities was reported. The trends of inequality were different in absolute and relative inequality stratifiers used, which needs for multilevel interventions to get unimmunized children for DPT3. Policies employing both demand- and supply-side interventions are required to reduce these inequalities. The government of Ethiopia needs to give due emphasis to poor, less educated, and rural women. Moreover, health policy has to be structured and implemented to address context-specific socioeconomic, infrastructural, and regional barriers of inequality among disadvantaged population sub-groups and sub-regions.

## Supporting information

**S1 Table. Community and individual level factors associated with DPT3 immunization using Multilevel Binary logistic regression in under–five children from 2000 to 2019.**
(DOCX)

**S1 File. DPT3 Stat EDHS for five surveys.**
(DTA)

## Acknowledgments

The authors would like to thank EDHS data base for sharing the data used in this analysis. We are also grateful to WHO for making the HEAT software available to us at no cost. We also acknowledge Addis Ababa University and individuals who were directly or indirectly involved in the preparation of this manuscript.

## Author Contributions

**Conceptualization:** Hailu Fekadu, Damen Hailemariam.

**Data curation:** Hailu Fekadu.

**Formal analysis:** Hailu Fekadu, Wubegzier Mekonnen, Aynalem Adugna, Helmut Kloos, Damen Hailemariam.

**Funding acquisition:** Hailu Fekadu.

**Investigation:** Hailu Fekadu, Wubegzier Mekonnen, Aynalem Adugna, Helmut Kloos, Damen Hailemariam.

**Methodology:** Hailu Fekadu.

**Project administration:** Hailu Fekadu.

**Writing – original draft:** Hailu Fekadu.

**Writing – review & editing:** Hailu Fekadu, Wubegzier Mekonnen, Aynalem Adugna, Helmut Kloos, Damen Hailemariam.

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
