## [Decision Letter · Decision Letter 0]

18 Jul 2023

PONE-D-23-05655Trends of Inequality in DPT3 Immunization Services Utilization in Ethiopia and its Determinant factors: Evidence from Ethiopian Demographic and Health Surveys, 2000- 2019PLOS ONE

Dear Dr. Fekadu,

Thank you for submitting your manuscript to PLOS ONE. After careful consideration, we feel that it has merit but does not fully meet PLOS ONE’s publication criteria as it currently stands. Therefore, we invite you to submit a revised version of the manuscript that addresses the points raised during the review process.

We look forward to receiving your revised manuscript.

Kind regards,

Pritam Ghosh

Academic Editor

PLOS ONE

Journal Requirements:

The fund is from Addis Ababa University additional support from Professor Helmut 

Reviewers' comments:

Reviewer's Responses to Questions

**Comments to the Author**

1. Is the manuscript technically sound, and do the data support the conclusions?

Reviewer #1: Yes

Reviewer #2: Partly

2. Has the statistical analysis been performed appropriately and rigorously? 

Reviewer #1: Yes

Reviewer #2: No

3. Have the authors made all data underlying the findings in their manuscript fully available?

Reviewer #1: Yes

Reviewer #2: No

4. Is the manuscript presented in an intelligible fashion and written in standard English?

Reviewer #1: Yes

Reviewer #2: Yes

5. Review Comments to the Author

Reviewer #1: Over a 20-year period, this study examined DPT3 immunisation service use in Ethiopia. Using several inequality indicators, the authors showed that national coverage increased, but disparities based on economic position, education, area, and other variables remained. The authors effectively emphasise the need to address discrepancies in immunisation coverage while acknowledging the various causes of these disparities. Overall, the write-up and analysis give an adequate foundation for the study and a clear framework for understanding the significance and relevance of the research issue.

ABSTRACT:

The abstract is well-written and structured. The background and method part is cleared and easy to understand. However, the authors may work on the result section with more structured sentences with proper flow.

INTRODUCTION:

The introduction heightened Global health initiatives, including the Global Vaccine Action Plan; role of DPT3 immunization coverage remains low in Ethiopia, despite efforts to improve it through strategies like the Reaching Every District approach.

Although the introduction section mentioned inequalities in immunization coverage based on socioeconomic, geographic, and demographic factors, a details discussion is needed with literature evidence for similar setups like Ethiopia.

METHOD:

The study setting and time, as well as the data sources, are effectively documented. The selection of the different inequality measurements is also well justified in the write-up.

However, mathematical formulae might aid the reader in better understanding the inequality measures.

RESULT:

The table need proper formatting. For example in Table 1 and Table 2, authors have used %(95%)UI. However, for general readers, it will be difficult to distinguished between Uncertainty interval (UI) and Confidence interval (CI). The term is new to me too. Are they meant same or different. If these two terms are different then please, the author need to elaborate details on Uncertainty interval (UI) in method section as well as in the tables as footnotes. If Uncertainty interval (UI) and Confidence interval (CI) same please justify why authors used Uncertainty interval (UI) instead of Confidence interval (CI) which is more conventional and accepted in the research.

Reviewer #2: Thank you for allowing me to review a manuscript entitled "Trends of Inequality in DPT3 Immunization Services Utilization in Ethiopia and its Determinant factors: Evidence from Ethiopian Demographic and Health Surveys, 2000- 2019" for your esteemed journal. However, after a careful review, I find the following points to share with you:

1. The introduction section is very general. The author may include a global overview of immunization, referring to developed and developing countries.

2. The literature review should also include the major challenges in utilizing immunization across the globe.

3. Research gap needs to be strengthened.

4. The study has used historical data from 2000 till 2019; therefore, it is important to mention the sample size and data compatibility across the various rounds.

5. The methods of inequality used in the study are very general and require methodological definitions for better clarification. Moreover, authors may think to use some advanced methods of inequality in spite of multiple methods, such as the Gini index etc.,

6. There should have been a sample summary table to understand the background characteristics of the sample of various rounds.

7. Although the higher wealth index group shows higher coverage in the utilization compared to the poorest group, the interpretation should also look into the fact that the level of utilization has increased at three times during the mentioned period among the poorest group, which is much higher than the highest quintile group.

8. There is a significant increase in the level of immunization among poor people, the authors should discuss the policy efforts and other dynamics for such improvement in the discussion section.

Dr Sanjit Sarkar

6. PLOS authors have the option to publish the peer review history of their article (what does this mean?). If published, this will include your full peer review and any attached files.

Reviewer #1: No

Reviewer #2: **Yes: **DR SANJIT SARKAR

<quillbot-extension-portal></quillbot-extension-portal>

---

## [Author Response · Author response to Decision Letter 0]

7 Sep 2023

Response to Review Comments 

Title: Trends of Inequality in DPT3 Immunization Services Utilization in Ethiopia and its Determinant factors: Evidence from Ethiopian Demographic and Health Surveys, 2000- 2019 Thank you for allowing me to review a manuscript entitled "Trends of Inequality in DPT3 Immunization Services Utilization in Ethiopia and its Determinant factors: Evidence from Ethiopian Demographic and Health Surveys, 2000- 2019" for your esteemed journal. However, after a careful review, I find the following points to share with you: 

1. The introduction section is very general. The author may include a global overview of immunization, referring to developed and developing countries.

#Response: Yes you are right, now it is updated and corrected based on your valuable comment

2. The literature review should also include the major challenges in utilizing immunization across the globe. 

#Response: Yes you are correct, now it is updated and major challenges in utilizing immunization across the globe is included based on your valuable comments

3. Research gap needs to be strengthened. 

#Response: Yes you are right, now it is updated and research gaps needs to be strengthen were included based on your valuable comment

4. The study has used historical data from 2000 till 2019; therefore, it is important to mention the sample size and data compatibility across the various rounds. 

#Response: Yes you are right, now the sample size and data compatibility across the various rounds was mentioned in the manuscript based on your valuable comment

5. The methods of inequality used in the study are very general and require methodological definitions for better clarification. Moreover, authors may think to use some advanced methods of inequality in spite of multiple methods, such as the Gini index etc.

#Response: Yes you are right, now methodological definitions were given based on your valuable comment and some advanced methods of data analysis have been done and shown in the manuscript and concentration curve is attached as Fig 7 with this revised manuscript

6. There should have been a sample summary table to understand the background characteristics of the sample of various rounds.

#Response: Yes you are right, now sample summary table has been included in the manuscript to show and to understand characteristics of the samples as indicated in your valuable comment

7. Although the higher wealth index group shows higher coverage in the utilization compared to the poorest group, the interpretation should also look into the fact that the level of utilization has increased at three times during the mentioned period among the poorest group, which is much higher than the highest quintile group.

#Response: Yes it is true, now it is updated and corrected based on your valuable comment

8. There is a significant increase in the level of immunization among poor people, the authors should discuss the policy efforts and other dynamics for such improvement in the discussion section. 

#Response: Yes you are right, now it is updated and some justification is given based on your valuable comment

9. Recommendation: This is an important study in the field but requires major revisions.

 Dr Sanjit Sarkar

Author : I thank you for your detail reviewing and your valuable comment

---

## [Decision Letter · Decision Letter 1]

11 Oct 2023

Trends of Inequality in DPT3 Immunization Services Utilization in Ethiopia and its Determinant factors: Evidence from Ethiopian Demographic and Health Surveys, 2000- 2019

PONE-D-23-05655R1

Dear Dr. Hailu Fekadu 

We’re pleased to inform you that your manuscript has been judged scientifically suitable for publication and will be formally accepted for publication once it meets all outstanding technical requirements.

Kind regards,

Dr. Pritam Ghosh

Academic Editor

PLOS ONE

Reviewers' comments:

Reviewer's Responses to Questions

**Comments to the Author**

1. If the authors have adequately addressed your comments raised in a previous round of review and you feel that this manuscript is now acceptable for publication, you may indicate that here to bypass the “Comments to the Author” section, enter your conflict of interest statement in the “Confidential to Editor” section, and submit your "Accept" recommendation.

Reviewer #1: All comments have been addressed

Reviewer #2: All comments have been addressed

2. Is the manuscript technically sound, and do the data support the conclusions?

Reviewer #1: Yes

Reviewer #2: Yes

3. Has the statistical analysis been performed appropriately and rigorously? 

Reviewer #1: Yes

Reviewer #2: Yes

4. Have the authors made all data underlying the findings in their manuscript fully available?

Reviewer #1: Yes

Reviewer #2: No

5. Is the manuscript presented in an intelligible fashion and written in standard English?

Reviewer #1: Yes

Reviewer #2: Yes

6. Review Comments to the Author

Reviewer #1: The authors have done substantial changes to match the journal's standard and the paper is suitable for the publication.

Reviewer #2: Thank you for the opportunity to review the revised version of the paper. The authors addressed all the quarries, clarifications, and changes raised in a very systematic manner. I congratulate all the authors for writing this manuscript and recommend that the journal editor consider the manuscript for publication after verifying the proofreading.

---

## [Editor Report · Acceptance letter]

19 Oct 2023

PONE-D-23-05655R1 

Trends of Inequality in DPT3 Immunization Services Utilization in Ethiopia and its Determinant factors: Evidence from Ethiopian Demographic and Health Surveys, 2000- 2019 

Dear Dr. Fekadu:

I'm pleased to inform you that your manuscript has been deemed suitable for publication in PLOS ONE. Congratulations! Your manuscript is now with our production department. 

Kind regards, 

on behalf of

Dr. Pritam Ghosh 

Academic Editor

PLOS ONE